# Event-Based Motion Capture System for Online Multi-Quadrotor Localization and Tracking

**DOI:** 10.3390/s22093240

**Published:** 2022-04-23

**Authors:** Craig Iaboni, Deepan Lobo, Ji-Won Choi, Pramod Abichandani

**Affiliations:** New Jersey Institute of Technology, 323 Dr Martin Luther King Jr Blvd, Newark, NJ 07102, USA; csi3@njit.edu (C.I.); dl394@njit.edu (D.L.); jc423@njit.edu (J.-W.C.)

**Keywords:** event-based cameras, motion capture systems, multi-quadrotor systems, object detection, YOLO, k-d tree, datasets for robotic vision, neural network, pose estimation, motion planning, motion coordination

## Abstract

Motion capture systems are crucial in developing multi-quadrotor systems due to their ability to provide fast and accurate ground truth measurements for tracking and control. This paper presents the implementation details and experimental validation of a relatively low-cost motion-capture system for multi-quadrotor motion planning using an event camera. The real-time, multi-quadrotor detection and tracking tasks are performed using a deep learning network You-Only-Look-Once (YOLOv5) and a *k*-dimensional (*k*-*d*) tree, respectively. An optimization-based decentralized motion planning algorithm is implemented to demonstrate the effectiveness of this motion capture system. Extensive experimental evaluations were performed to (1) compare the performance of four deep-learning algorithms for high-speed multi-quadrotor detection on event-based data, (2) study precision, recall, and F1 scores as functions of lighting conditions and camera motion, and (3) investigate the scalability of this system as a function of the number of quadrotors flying in the arena. Comparative analysis of the deep learning algorithms on a consumer-grade GPU demonstrates a 4.8× to 12× sampling/inference rate advantage that YOLOv5 provides over representative one- and two-stage detectors and a 1.14× advantage over YOLOv4. In terms of precision and recall, YOLOv5 performed 15% to 18% and 27% to 41% better than representative state-of-the-art deep learning networks. Graceful detection and tracking performance degradation was observed in the face of progressively darker ambient light conditions. Despite severe camera motion, YOLOv5 precision and recall values of 94% and 98% were achieved, respectively. Finally, experiments involving up to six indoor quadrotors demonstrated the scalability of this approach. This paper also presents the first open-source event camera dataset in the literature, featuring over 10,000 fully annotated images of multiple quadrotors operating in indoor and outdoor environments.

## 1. Introduction

Code and Video: Github Link: https://github.com/radlab-sketch/event-quadrotor-mocap (accessed on 18 March 2022).

Multi-quadrotor systems are susceptible to various operational challenges such as sensitivity to external disturbances, increased computational cost due to scalability, inter-robot collision avoidance, and obstacle avoidance [1]. Several works that investigate the multi-quadrotor motion problem rely on infrared motion capture systems such as VICON [2,3,4] and OptiTrack [5,6,7] that provide ground truth position estimates to evaluate the performance of their algorithms. While such motion capture systems provide fast and accurate position, velocity, and pose estimates, they are expensive and marker-based and require powerful computing infrastructure for data acquisition and processing. As such, these motion capture systems are not easily accessible to robotics researchers [8].

The study presented in this paper investigates event-based cameras as an alternative to infrared motion capture systems. Specifically, this motion capture system uses an event camera and a state-of-the-art YOLOv5 deep learning-based, Convolutional Neural Network (CNN) to provide real-time estimates of the position and tracks of multiple quadrotors. Event-based cameras, also known as dynamic vision sensors (DVS), are neuromorphic sensors that record changes in light intensity asynchronously. These sensors are attractive to computer vision researchers and practitioners due to high-speed vision, low perception latency, and relatively low power requirements when compared to frame-based cameras [9,10,11,12].

**Need for Event Datasets:** The growing interest in event-camera-based vision tasks has created a need for labeled event-camera datasets that are suitable for evaluating event-vision tasks [13,14,15,16]. While event-based data that have been collected with the help of quadrotors are available online, to the best of our knowledge, there are no open-source event-based data that capture flying quadrotors in indoor and outdoor environments using fixed and moving cameras. Through this paper, we are open-sourcing the first event dataset of multiple (up to six) quadrotors flying simultaneously in an indoor and an outdoor environment. The dataset is available here [17].

The key contributions of this study are as follows:**An event-based motion capture system for online multi-quadrotor motion planning**. The implementation details and hardware–software architecture have been detailed in this manuscript. Comprehensive experimental validation results are provided to showcase demanding use-cases beyond a standard indoor laboratory setup. Additionally, key software functionality has been open-sourced.**A novel open-source event camera dataset of multiple quadrotors flying simultaneously at various speeds and heights.** To the best of our knowledge, this dataset is the first to involve event cameras in the capture of quadrotors in flight. Over 10,000 synthetic event images were captured over a series of flight test sessions taking place in both indoor and outdoor arenas. Challenging scenarios, including low-light conditions and unstable camera motions, are represented in the dataset. Bounding box annotations for each image are provided.

## 2. Related Work

Several studies in the literature have used Vicon and OptiTrack infrared-based motion capture systems for multi-quadrotor experiments [3,18,19,20].

The event-based motion capture system presented here draws its working principles from the computer vision tasks of event-based multi-object detection (MOD) and multi-object tracking (MOT). A relatively small yet growing body of work underscores the value of event-based cameras for multi-object detection and tracking [16,21,22,23,24,25,26]. Approaches to perform event-based detection include clustering event data [16], using sliding window techniques [22], using event stream representations that approximate the 3D geometry of the event stream with a parametric model [23], and performing multi-level fusion using novel representations of event data (Frequency, Surface of Active Event (SAE), and Leaky Integrate-and-Fire (LIF)) [25].

In particular, leveraging deep learning methods for event-based work combines the power of cutting-edge vision research and high-frequency event sensing [27]. For example, by using regions-of-interest in CNNs for frame-based and event-based sensors, the authors in [24] were able to successfully build a tracking system that was evaluated on the Ulster dataset that featured moving objects in the cluttered background with ego-motion. The results showed 90% tracking accuracy with 20 pixel precision. CNNs can also be used to create more sophisticated event-data systems, as exemplified by the authors in [28]. Here, the authors presented a detection and tracking framework featuring a CNN-based offline-trained detector and a complimentary filter-based online-trained tracker. Subsequently, a Kalman filter-based fusion scheme maintained robust tracking for real-time operations.

Similarly, in [29], the authors used multiple CNNs and event data to perform fast pedestrian detection. Subsequently, the authors implemented a confidence map fusion method combining the CNN-based detection results to obtain higher accuracy. The use of CNNs has been extended to performing nuanced event-based computer vision tasks as witnessed in [30]. The authors implemented an event-camera-based driver monitoring system (DMS) that performed real-time face and eye tracking, and blink detection. In [31], the authors evaluated 3D object tracking using event data, with and without prior image reconstruction using a combination of the Channel and Spatial Reliability Tracking (CSRT) algorithm and a CNN for depth estimation. The above works provide evidence of the natural confluence between deep learning and event data. Our study extends the above body of work by leveraging this confluence to benefit the robotics research community.

We note that there is a growing body of literature on spiking neural networks (SNNs), which have shown potential to take advantage of the asynchronous nature of event data and low latency of event cameras [32]. However, the absence of a region proposal network and limited access to commercial-off-the-shelf (COTS) neuromorphic computing hardware makes SNNs for motion capture a subject of further investigation [33].

Most of the works mentioned above make use of open-source datasets for training the neural networks [34,35,36]. In a similar vein, we present the first open-source event dataset to assist detection and tracking tasks featuring multiple quadrotors operating under various lighting and environmental conditions. This dataset was not simulated or converted from RGB-based frames but was instead captured from real-world flight scenarios. Additionally, unlike other event datasets that have been generated by moving the event camera in front of a standard display [36], the dataset presented in this paper captured the true motion of quadrotors in real-world indoor and outdoor conditions. This dataset adds to the growing body of open-source datasets that benefit the event camera research and practitioner community [36,37].

The remainder of this paper is organized as follows. In Section 3, the hardware and software architecture of the motion capture system is described. Section 4 covers the results of the experiments in-depth. Section 5 provides a pointed discussion about the strengths and limitations of this study, and Section 6 concludes the paper with some future directions.

## 3. System Architecture

This section elucidates the hardware and software setup for capturing real-time event information from the camera and setting up the motion-capture process.

### 3.1. Hardware

The hardware consisted of the event camera, the quadrotors, a host computer, and a Google Colaboratory cloud computing environment.

#### 3.1.1. Camera Specifications

The event camera used in the experiments was a VGA-resolution contrast-detection vision sensor from Prophesee, shown in Figure 1. This camera features a CMOS vision sensor with a resolution of 640×480 (VGA) pixels with 15μm ×15μm event-based pixels and a high dynamic range (HDR) beyond 120 dB. The camera ran on 1.8 V supplied via USB, with a 10 mW power dissipation rating in low power mode. The camera interface was established using USB for communication and was mounted on the ceiling looking down at the experiment area as depicted in Figure 1.

#### 3.1.2. Quadrotor Specifications

The quadrotor used in this study was an off-the-shelf, Python programmable, small unmanned aerial vehicle (sUAV). The quadrotor’s inertial measurement unit (IMU) featured an accelerometer, gyroscope, barometer, and magnetometer for flight control. The quadrotor flew at a maximum speed of 8 m/s [38]. Communication between the quadrotor and the host computer occurred over a WiFi network connection. The quadrotors were controlled via Python commands that specified body-frame velocity over the WiFi network. The quadrotor was rated to process radio control commands at 1000 Hz.

#### 3.1.3. Host Computer

The host machine was responsible for interfacing with the event camera, performing quadrotor detection and tracking, and executing the optimization-based motion planning algorithm. The host computer was equipped with the Windows 10 Operating System, an NVIDIA GTX 1080 GPU, Intel i7 CPU, and 16 GB RAM.

#### 3.1.4. Google Colaboratory (Colab) Cloud Computing Environment

The Colab environment was responsible for offline training and benchmarking the deep learning networks used in this work. Colab is a hosted Jupyter notebook service that requires no setup to use while providing access to high-performance GPUs [39]. The cloud computing environment in this work leveraged the NVIDIA Tesla K80 GPU.

### 3.2. Software

The software was developed in Python 3.7 according to a modular, multi-service architecture as shown in Figure 2. The software architecture consisted of two key software services: (1) detection and tracking and (2) motion planner. The connections between services consisted of inter-process, intra-process, and WiFi connections. The inter-process connection allowed for exchange of data via Address Event Representation (AER) and User Datagram Protocol (UDP). The intra-process connections shared data via in-process memory (such as program variables), and WiFi was used to broadcast motion commands to multiple quadrotors.

#### 3.2.1. Detection and Tracking Service

The detection and tracking service was responsible for event data processing, performing quadrotor detection and classification, and sending tracking information to the motion planning service.

##### Event Accumulator

The event camera communication used a set of data acquisition functionalities (API calls) provided by the manufacturer to read event data.

The event data contained information about the location of a pixel ($e_{x}$, $e_{y}$) where the event occurred, the event polarity $e_{+/-}$ (positive or negative), and the timestamp for the event $e_{t}$. An event’s polarity was positive when there was a positive change in light intensity, and it was negative when there was a negative change in light intensity. A data buffer continuously collected event objects from the camera’s output stream. This information was then encoded into address-events that were asynchronously transmitted (ultimately as a synthetic frame) at the periphery via the AER protocol. This process was repeated at 10 KHz.

The parameter accumulation time $t_{a}$ was used to specify the time in seconds for which events were accumulated into each synthetic frame from the past to the current time. The effect of $t_{a}$ is presented in Figure 3. $t_{a}$ = 0.1 s showed the best results and was used for subsequent experiments. Readers are referred to [16] for further details about the effect of accumulation time $t_{a}$ value on synthetic frame quality.

Each synthetic frame was aggregated from events in a spatio-temporal window W with $t_{a}$ specifying the length of this temporal window. At the start of each temporal window generation process, an empty 2D grid was formed with dimensions 640 pixels × 480 pixels. This grid size matches the resolution of the event camera sensor. Pixels on the event camera sensor that experienced event activity were inserted at the corresponding coordinates on the 2D grid as positive or negative polarity values. When the timestamp of an event exceeded the length of the temporal window ($t_{a}$), the event accumulator would pass all events in W to the CNN as a synthetic frame. Subsequently, a new temporal window was generated and the process was repeated.

##### Quadrotor Detection

Quadrotor detection was based on YOLOv5 [40]. YOLO is a family of single-shot detectors that detect and classify images in a single step [41]. Due to YOLO’s rapid inference speeds, it is well-suited for real-time applications and was the chosen detection method in this work. The YOLOv5 algorithm was trained on event data footage of the quadrotors to detect and classify target quadrotors. Detailed information on the training is provided in Section 4.1. The trained YOLOv5 network was then used to report on the contents of synthetic event frames by performing detection and classification of multiple quadrotors. Thus, YOLO provided real-time ($x^{i}$, $y^{i}$) locations of the center points $C^{i}$ of each moving quadrotor *i* in the scene at any given time *t*. Inference with YOLO outputs a class, confidence score, and bounding box location for each detection. Each quadrotor’s location was passed to the tracking service if detection was made successfully.

##### IDTrack: Tracking Service

Tracking quadrotors in real time is an essential step to bridge the gap between detections and motion planning. A *k*-dimensional (*k*-*d*)tree was used for tracking due to the tree structure’s efficiency for nearest-neighbor queries [16,42]. The purpose of the tracking service was to assign and maintain persistent identifiers (IDs) for detected quadrotors.

The event capture suffered from spurious noise effects that led to new IDs being assigned to quadrotor-like objects (such as quadrotor reflections on the ground). Additionally, when quadrotors would slow down or hover in place, the event camera reported fewer events (and generated indistinct images), causing lost tracks/IDs or mislabeling of quadrotors. These issues were addressed by developing a process called IDTrack. IDTrack leveraged nearest neighbor searches between quadrotor detection central points. Assigned quadrotor IDs were stored with the last known (*x*, *y*, t−1) position in a *k*-*d* tree to keep the quadrotor positions over time, even in scenarios where detections were not possible.

IDTrack was initiated as soon as information about the first detected quadrotor central point $C^{i=1}$ was added to the empty *k*-*d* tree τ(0).

After this initiation step, a nearest neighbor search τ(t).NN was conducted for all other detected central points $C^{j}\left(t\right)$ at a given time step *t*. The Euclidian distance $d^{ij}\left(t\right)$ between $C^{j}\left(t\right)$ and its nearest neighbor $C^{i}\left(t\right)\in \tau \left(t\right)$, ∀i,j∈{1,…n},i≠j was used to define two possible cases:**New Quad Discovered:** If this distance $d^{ij}\left(t\right)$ was greater than the width of the physical quadrotor chassis σ, the new detection was inferred as a distinct quadrotor. A new node corresponding to this newly discovered quadrotor was created in τ(t) with ($C^{j}x\left(t\right),C^{j}y\left(t\right)$) coordinates. A counter variable called nextID was incremented by one each time a new node corresponding to a distinct quadrotor was added to τ(t). This counter helped keep track of the sequence of IDs being assigned to the newly discovered detection central points.**Same Quad Rediscovered:** On the other hand, if $d^{ij}\left(t\right)$ was less than or equal to σ, the central point $C^{j}\left(t\right)$ was inferred to belong to the same quadrotor represented by central point $C^{i}\left(t\right)\in \tau$. In this case, $C^{i}\left(t\right)\in \tau$ was overwritten by $C^{j}\left(t\right)$ as the latest central point information about the corresponding quadrotor.

The known location, along with respective IDs, of each quadrotor *i* was passed to the motion planner ($x^{i}$, $y^{i}$, $t^{i}$, ID${}^{i}$).

#### 3.2.2. Motion Planner Service

The motion planner service was responsible for determining and ascertaining motion commands to guide the quadrotors to their destination while avoiding collisions. The motion planner service was created in a modular fashion such that most motion planning algorithms could be interfaced with through standard API calls. For this study, a decentralized optimization-based motion planning algorithm was used. This motion planner used a receding horizon (RH) architecture and ensured that each quadrotor could calculate its position and velocity while maintaining a collision avoidance distance of at least $d_{safe}$ cm with other quadrotors. Given the decentralized nature of the motion planner, each quadrotor was accorded its own instance of the motion planner in software architecture and implementation. More information about this mixed-integer nonlinear programming (MINLP)-based motion planner can be found in our previous work [43]. Given that most quadrotors experience some level of drift in their motion, a **flight safety corridor** was defined around each waypoint. The motion planner commanded each quadrotor to stay within this predefined flight safety corridor as it traversed the waypoints. These flight corridors are depicted in Figure 4.

As depicted in Figure 2, the optimization-based motion planner received updated positions of the quadrotors from the tracking service. It generated linear and angular velocity commands that were resolved into the appropriate roll, pitch, throttle, and yaw to be specified at each time *t* for each quadrotor *i*. The host machine sent these motion commands to each quadrotor through a WiFi connection. The linear and angular velocity commands were appropriately constrained (<80%) such that the motors of the quadrotors were never saturated. The roll, pitch, and yaw commands were implemented by the onboard flight controller of the quadrotor.

The detection and tracking service and the motion planning service were asynchronous, i.e., the detection and tracking service reported new positions as fast as possible, and the motion planning service requested the latest positions as and when it needed it. A quadrotor was instructed to land if the list of waypoints was empty, meaning it had completed its mission.

## 4. Experimental Setup and Results


*Note: Readers are referred to the accompanying video for footage of the flight tests. Relevant Python functionalities have been open-sourced for the community at Github Link: https://github.com/radlab-sketch/event-quadrotor-mocap (accessed on 18 March 2022)*


Experiments performed in this work are grouped into ’offline’ and ’online’ categories determined by the GPU used to perform the computation. Offline experiments performed on the NVIDIA Tesla K80 GPU provided by the Google Colab environment were conducted to measure the speed and effectiveness of the CNNs. Online experiments performed on the host machine’s NVIDIA GTX 1080 GPU were conducted to study the practical performance of the motion capture system.

The event camera was mounted on the ceiling directly above a 183 × 183 cm area. The area was covered with white foam mats to suppress light reflections on the glossy floor. For flights in outdoor environments, foam mats were not used. All quadrotor tests were performed at 80 cm AGL (above ground level). Quadrotors were instructed to follow a series of waypoints in one of four patterns, as illustrated in Figure 4: square, circle, spline, and lawnmower shape. An ordered collection of waypoints was maintained for each quadrotor, forming the 2D path shapes when viewed above. Additionally, scenarios were constructed to demonstrate collision avoidance with two and three active quadrotors at varying detection ranges. In all experiments, quadrotors flew autonomously across the sensor’s field of view while ensuring that each goal location was reached. The flight tests lasted between 30 s and 120 s. Flight tests were repeated five times for each experimental category.

Figure 5 depicts the three categories of experimental evaluations that were performed. These are (1) comparison of multiple modern CNNs for detection of multiple airborne quadrotors, (2) robustness of the system to varying light conditions and camera motions, and (3) system performance validation for motion coordination of multiple quadrotors.

The key metrics used in the study were divided into two categories as explained in the following:**Detection Metrics:** The detection performance was assessed using Precision P, Recall R, and F1 score F. Precision is defined as the ratio of the number of correct detections to the total number of detections. Recall is defined as the ratio of the number of correct detections to the total number of true objects in the data.
(1)$$\mathrm{Precision}\;P=\frac{TP}{TP+FP}$$
(2)$$\mathrm{Recall}\;R=\frac{TP}{TP+FN}$$
(3)$$F1\;\mathrm{Score}\;F=2\times \frac{P\times R}{P+R}$$TP, FP, and FN were the number of True Positives, False Positives, and False Negatives, respectively.**Motion Planning Metrics:** The motion planner’s performance was captured using two metrics: (1) waypoint navigation task via waypoint boundary violation rate O and (2) the duration of each flight test $T_{mission}$.O was the ratio of the number of frames that the quadrotor was out of bounds of the flight safety corridor containing the waypoints over the number of elapsed frames for the experimental run. O was calculated as follows:
(4)$$O=\frac{\mathrm{Number}\;\mathrm{of}\;\mathrm{out}\;\mathrm{of}\;\mathrm{bounds}\;\mathrm{frames}}{\mathrm{Total}\;\mathrm{number}\;\mathrm{of}\;\mathrm{frames}}$$$T_{mission}$ was the time it took for the last quadrotor to land during each flight test in seconds.O and $T_{mission}$ calculations began after the quadrotors were airborne and instructed to proceed to the first waypoint.

Table 1 provides a map of the evaluation dimension, method, and metric for all experiments.

### 4.1. Dataset

A key contribution of this work was the event dataset that involved multiple quadrotors flying in indoor and outdoor arenas under various lighting conditions [17]. The dataset consists of synthetic frames formed from a collection of events obtained through extensive flight tests. Approximately 10,000 images of between one and three quadrotors flying at speeds ranging from 0.2 m/s to 1 m/s and heights ranging from 0.3 m to 1 m were labeled using an annotation tool by human experts. Frames are provided in 640 × 480 pixel resolution. The novel dataset has been made available at our Github repository found in [17]. Specifically, the dataset involves:8300 (83%) images that depict quadrotors flying variable paths in an indoor arena;500 (5%) images that depict quadrotors flying variable paths indoors using an unstable camera;500 (5%) images that depict quadrotors flying variable paths indoors under low light conditions;700 (7%) images that depict quadrotors flying variable paths outdoors using an aerial camera.

Both low- and high-speed maneuvers were meticulously represented to ensure that sparse and dense event activity would result in valid detections. The input dataset was divided into three subsets, with 70% of the total dataset allocated to the training set, 20% for validation, and 10% for the testing set.

### 4.2. Training Results of YOLOv5

Combinations of data augmentations such as rotations, distortions, mixup, cutout, and occlusions were randomly applied to the training dataset to train the YOLOv5 network. A learning rate of $1\times 10^{-2}$ with a learning rate decay of 0.999 was used. The batch size used for training was 16 images per batch. The training occurred over 125 epochs. The optimizer is used as Stochastic Gradient Descent (SGD) with a momentum of 0.937 and weight decay of $5\times 10^{-4}$. The training process used the PyTorch framework and took place on the Google Colaboratory cloud environment [39,44]. The results of the training procedure, evaluated at each epoch in terms of precision, recall, and F1 score, are shown in Figure 6. As observed, the training results showed favorable convergence rates, with evaluations occurring on the validation set after each epoch. The training was halted after 125 epochs to prevent overfitting.

### 4.3. Comparative Analysis with State of the Art

An expansive set of experiments and associated analysis was performed to compare the multi-quadrotor detection and run-time performance of four modern neural networks: YOLOv5 [40], YOLOv4 [41], Faster R-CNN [45], and RetinaNet [46]. Key insights are derived at the end of several experiments presented in this section. The Faster R-CNN network is a prominent two-stage object detector consisting of a Region Proposal Network (RPN) and a Fast R-CNN detector. RetinaNet is a one-stage object detector that predicts bounding boxes in a single step without region proposals. RetinaNet is notable for competing with the accuracy of two-stage network architectures while yielding faster detections.

#### 4.3.1. Detection Performance

Detection results for experiments with YOLOv4, YOLOv5, Faster R-CNN, and RetinaNet on detection metrics are noted in Figure 7. It is observed that YOLOv5 and YOLOv4 demonstrated similar performance, with YOLOv5 marginally outperforming YOLOv4 in terms of precision value. The YOLO algorithms outperformed Faster R-CNN and RetinaNet in terms of precision and recall. YOLOv5 outperformed YOLOv4 with respect to precision, recall, and sampling rate. Additionally, the benefits of the YOLOv5 PyTorch implementation over the YOLOv4 Darknet implementation include fast and seamless integration with the Python-based motion planners. In this context, we have considered YOLOv5 as the CNN of choice for this study.

**Key Insight:** Of note is the fact that, for small-form factor-embedded devices running object detection tasks, the size of the network weights should be considered when choosing an appropriate algorithm for detection applications. Faster R-CNN produced a weights file of 314.7 MB size, and RetinaNet produced a weights file of 120 MB size. Note that YOLOv5 produced a smaller weights file than the compared object detection networks, at 14 MB, while YOLOv4 produced a weights file of 22 MB size.

#### 4.3.2. Sampling (or Inference) Rate Analysis

Additionally, single image sampling (or inference) rates for all algorithms were evaluated and are depicted in Figure 8. Sampling/inference rate evaluations were run on both Google Colaboratory (using NVIDIA Tesla K80 GPU) and the host machine (using NVIDIA GeForce 1080 GPU) and are noted for both platforms in Figure 8. Similar sampling/inference rate results were observed on both hardware platforms and YOLO models, with YOLOv5 operating faster than YOLOv4 in all scenarios. However, both YOLO models outperformed the Faster R-CNN and RetinaNet models.

On the host machine (GTX 1080), YOLOv5 performed 1.14× faster than YOLOv4, 4.8× faster than Faster R-CNN, and 12× faster than RetinaNet. On the Google Colab environment (Tesla K80), YOLOv5 performed 1.04× faster than YOLOv4, 6.14× faster than Faster R-CNN, and 13.5× faster than RetinaNet.

As expected from a larger GPU with more processing power, the NVIDIA Tesla K80 provided by Google Colaboratory was capable of greater sampling rates than the host machine’s NVIDIA GTX 1080.

**Key Insight:** Faster R-CNN performed the inference task slower than the YOLO variants due to more complex architecture. Additionally, as expected from a larger GPU with more processing power, the NVIDIA Tesla K80 provided by Google Colaboratory was capable of greater sampling rates than the host machine’s NVIDIA GTX 1080.

### 4.4. Robustness Analysis

Several experiments that involved flight tests were conducted to characterize the robustness of the motion-capture system to unfavorable ambient light conditions and event camera motion.

#### 4.4.1. Effect of Varying Ambient Lighting Conditions

As event-based cameras report brightness changes detected by each pixel, the performance of the camera is subject to environmental lighting conditions [47]. Two ambient lighting settings were created by using LED light strips modulated to provide strong and weak lighting intensities. A Lux Meter was used to quantify the amounts of light present in these ambient lighting conditions. The two conditions featured the use of LED lights at strong intensity (652 Lux) and the use of LED lights at weak intensity (23.42 Lux), with all other lighting turned off. These two lighting conditions are shown in Figure 9. These conditions have been chosen to be representative of commonly encountered real-world lighting conditions. Environments with strong intensity lighting represent the condition of flying outdoors during daytime hours or flying in well-illuminated indoor environments. Weak intensity lighting represents the condition of flying outdoors at dusk or within dimly lit indoor environments. In environments with weak intensity lighting (23.42 Lux), precision and recall values of 0.96 and 0.97, respectively, were recorded for YOLOv5. These results indicate that there was a slight degradation of detection performance under weak lighting conditions.

**Key Insight:** Event-based cameras are sensitive to small changes in brightness, which proved sufficient to capture motion in the scene. Event-based detections prevailed in the face of low-light conditions. It should be noted that despite the reliable performance noted in varied lighting conditions, some external lighting is required for the camera to report changes in brightness. Event-camera-based systems usually fail to detect objects at night or in total darkness.

#### 4.4.2. Effect of an Unstable Camera on Detection Performance

Two experimental conditions were used to evaluate the effect of an unstable camera.

In one case, a rope was affixed to the platform and harness attaching the event camera to the ceiling of the indoor experiment area. The rope was constantly pulled, introducing motion to the camera that could be observed as noise artifacts in the event frame. Readers are referred to the accompanying video for relevant footage.In the second case, the event-based camera was affixed to a quadrotor in a downward-facing orientation. Although the camera-equipped quadrotor was flown above the motion-coordinated quadrotors and instructed to hold the position in space, drift due to wind introduced instability to the quadrotor and camera.

Additionally, quadrotor positions relative to the camera frame were subject to variation depending on the intensity of camera motion. In indoor scenarios with an unstable camera, precision and recall values of 0.94 and 0.98, respectively, were observed with YOLOv5. In outdoor operation scenarios, precision and recall values of 0.95 and 0.98 were achieved, respectively, with YOLOv5.

Although quadrotor positions in frames fluctuated with camera motion, detection results were accurate within a range of 5 cm.

**Key Insight:** Due to the microsecond temporal resolution of event-based cameras, minimal motion blur was observed during scenarios with an unstable camera, thus preserving the consistent appearance of quadrotors regardless of camera stability. Therefore, the main concern with unstable cameras was addressing the distance between any given quadrotor from one frame to the next within the tracking method. This is because IDTrack’s nearest neighbor search must assimilate a quadrotor’s position from one frame to the next to facilitate successful tracking across frames, despite a potential detection at up to 5 cm from the quadrotor’s previous position. Therefore, for a given quadrotor, the minimum Euclidean distance between successive detections should be sufficiently large so as to prevent the erroneous assignment of a new ID.

### 4.5. Performance Validation for Motion Coordination

These experiments focused on studying (1) the effect of varying $d_{safe}$ between quadrotors on $T_{mission}$, (2) the effect of increasing the $n_{quad}$ on $T_{mission}$, and (3) the effect of varying quadrotor paths on O. Each flight test in these experiments was repeated five times, and the average results across these five tests are reported in the following section.

#### 4.5.1. Effect of $d_{safe}$ on $T_{mission}$

Experiments were performed at 5 cm $\leq d_{safe}\leq$ 20 cm and 20 cm $\leq d_{safe}\leq$ 55 cm. Flight tests for these experiments involved two and three quadrotors set to fly towards one another as depicted in Figure 10. The motion planner was successfully able to ensure that no $d_{safe}$ violations occurred. From Table 2, it is observed that $T_{mission}$ was inversely related to $d_{safe}$. Readers can also view the footage for these experiments in the accompanying video.

#### 4.5.2. Effect of Increasing $n_{quad}$ on $T_{mission}$

Figure 11 depicts flight tests with two, three, and six quadrotors. The effect of increasing $n_{quad}$ on $T_{mission}$ is reported in Table 2. It was observed that as $n_{quad}$ increased from 2 to 6 (with a $d_{safe}$ between 5 cm and 20 cm), $T_{mission}$ increased from 11 s to 19 s. This increase in $T_{mission}$ with increasing $n_{quad}$ is consistent with the expected increase in computational times of a decentralized motion planner. Of key note is the fact that the event-camera-based motion capture system was able to provide the required positioning information for performing successful motion planning and coordination.

#### 4.5.3. Effect of Varying Quadrotor Paths on Flight Corridor Boundary Violations O

Flight tests featuring one or two quadrotors operating in indoor and outdoor arenas were conducted to study the effect of varying quadrotor paths on O. The flight corridor boundary violations occur due to drift and imperfections in the quadrotor flight controls. Figure 4 depicts flight tests with square, circle, lawnmower, and cubic spline shaped quadrotor paths. Figure 12 depicts a two-quadrotor experiment in an outdoor setting. Table 3 reports the average O values for these flight tests. It was observed that O remained uniformly low across all shapes, indicating that the motion capture system was able to provide fast and accurate positioning information to the optimization-based motion planner for quadrotor course corrections. Readers can also view the footage for these experiments in the accompanying video.

## 5. Discussion

This study aimed to investigate whether event cameras and modern CNNs can provide suitable motion capture feedback for motion coordination of multi-quadrotor systems. The investigations provide reasonable evidence that event cameras can indeed provide a suitable motion tracking system for multi-quadrotor systems. In the following sub-sections, the strengths and limitations of this study are discussed.

### 5.1. Strengths

Experimental results indicate that YOLOv5 and YOLOv4 models performed well on GPU hardware for multi-quadrotor detection. YOLO architectures tested in this study outperformed two other comparable state-of-the-art CNN detector architectures. The performance of the proposed system displayed minimal deterioration under constrained lighting or camera instability.YOLOv5 ran natively on the Pytorch framework, making for a seamless development experience when interfacing with additional Python services. While YOLOv5 may be easier to bring quickly into production, YOLOv4 will still be used in circumstances where development in C using the Darknet neural network framework [48] is desirable.Detection performance remained consistently high across $n_{quad}$ representations, indicating that the method accommodates greater numbers of quadrotors.Runtime performance was unaffected by varying $n_{quad}$ value, supporting the notion that the approach is scalable beyond 6 active quadrotors in the arena.The resulting dataset from this study fills a much-needed gap in the aerial robotics community that is looking to work with event cameras.

### 5.2. Limitations

The minimum resolvable control distance for the quadrotor flight controller was 5 cm in the *x* and *y* directions. As such, it was not possible to test for scenarios with $d_{safe}\leq$ 5 cm.As this method of detection and tracking occurs in 2D space at a fixed perspective, there were some notable challenges. For example, quadrotors that flew close to the camera occupied a significant portion of the camera’s field of view and occluded other quadrotor activity. Taller ceilings or incorporating multiple camera sources to this method would expand the detection and tracking area of the indoor arena. Multiple event camera streams would allow quadrotors to follow three-dimensional paths within the experiment arena and introduce camera redundancy and resilience to occluded sensors.As depicted in Figure 12, outdoor experiments were conducted with two quadrotors. However, further outdoor experiments with a higher number of quadrotors would shed light on the performance of such a system in windy conditions.Finally, with any supervised CNN approach, there is a need for sufficient training data, which can be a time-consuming (and hence expensive) process. The open-sourced dataset accompanying study will provide a strong starting point for the research community.

## 6. Conclusions

This paper developed and experimentally validated an event-camera-based real-time motion-capture system for multi-quadrotor motion planning. Indoor and outdoor flight tests were performed to study the system’s performance. Multiple CNN detectors were evaluated in terms of precision, recall, and algorithmic runtime, with YOLOv5 proving to be the most effective architecture overall. The effect of a varying number of quadrotors, quadrotor paths, and collision safety distances on mission time and safety boundary violations was reported. The experimental results demonstrate the effectiveness of event cameras and neural-network-based object detection for multi-vehicle motion planning applications. Future work will extend this study to track and control multiple quadrotors in three-dimensional space using an asynchronous event stream. Additionally, quadrotor-mounted event cameras will be used to expand this work to a moving platform and follow the research approaches put forth in [49,50,51].

## Figures and Tables

**Figure 1 sensors-22-03240-f001:**
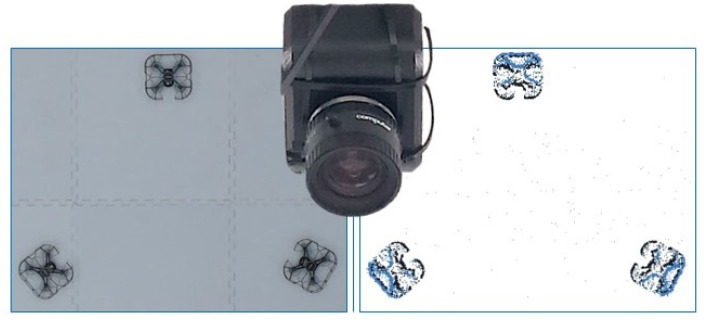
The event camera was mounted on the ceiling facing downward. Depicted here are 3 quadrotors positioned within the indoor arena, depicted in RGB (**left**) and event-based (**right**) camera streams.

**Figure 2 sensors-22-03240-f002:**
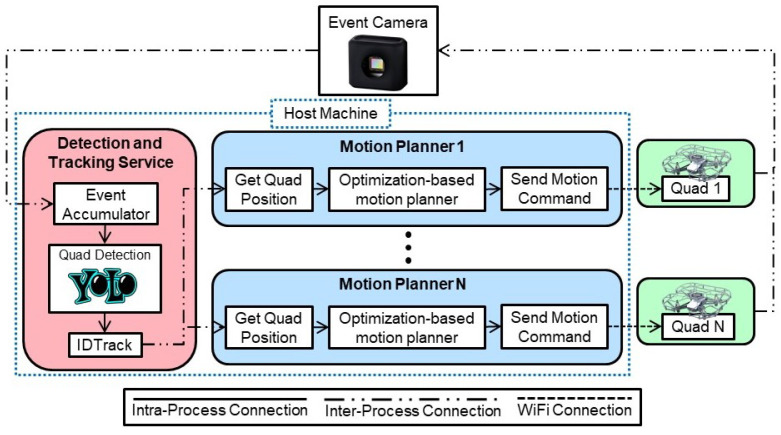
Modular multi-process system architecture developed as part of this study featured detection and tracking, and motion planning software services operated as processes on the host machine. The quadrotors received commands from their respective decentralized optimization-based motion planners over WiFi connections.

**Figure 3 sensors-22-03240-f003:**
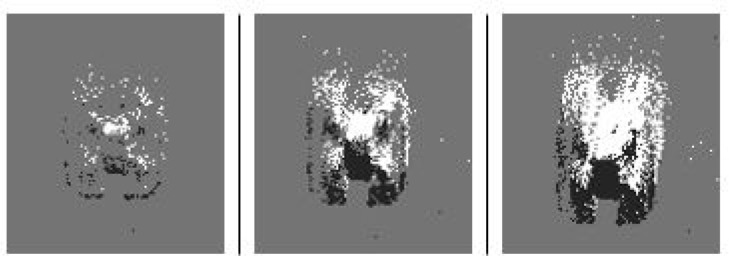
Various $t_{a}$ were assessed by training experimental networks before selecting a value for dataset capture. (Left to Right) $t_{a}$ = 0.05 s, $t_{a}$ = 0.1 s, and $t_{a}$ = 0.15 s. The $t_{a}$ that provided the values of precision and recall closest to 1 were used for all subsequent data and experiments.

**Figure 4 sensors-22-03240-f004:**
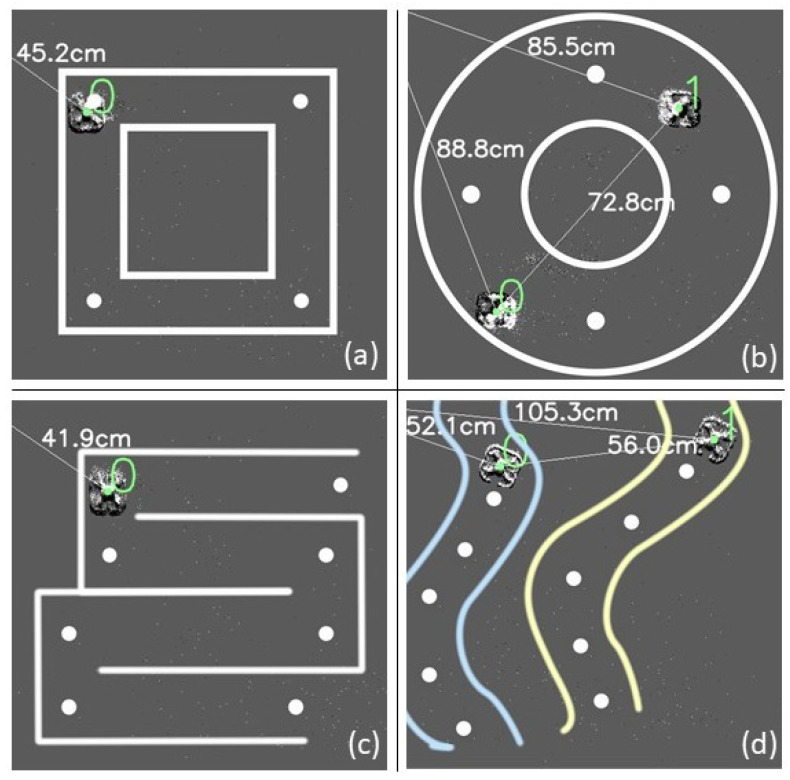
Path planning experiment scenarios with multiple quadrotors viewed from above. (**a**) Square path, (**b**) circle path, (**c**) lawnmower path, and (**d**) cubic spline path. A flight safety corridor (as indicated by solid lines) was defined to enclosed all waypoints. Each quadrotor was commanded by the motion planner to stay within this predefined flight corridor as it traversed the waypoints.

**Figure 5 sensors-22-03240-f005:**
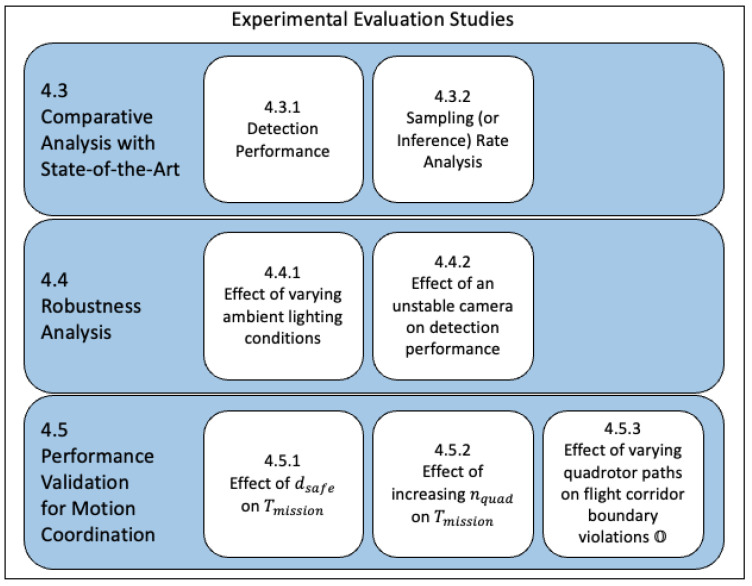
Three categories of experimental evaluations are presented in Section 4. These categories focus on CNN algorithm comparisons, robustness of the motion capture system to environmental conditions, and actual performance evaluation for multiple quadrotors motion planning.

**Figure 6 sensors-22-03240-f006:**
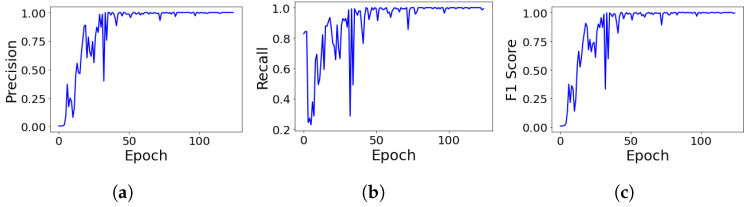
(**a**) Precision, (**b**) recall, (**c**) F1 score evaluated on the YOLOv5 validation set.

**Figure 7 sensors-22-03240-f007:**
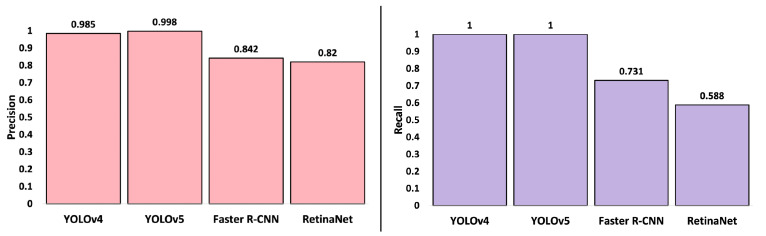
Performance metrics for a variety of object detection networks. Two YOLO architectures as well as the Faster R-CNN and RetinaNet methods are compared in terms of precision and recall.

**Figure 8 sensors-22-03240-f008:**
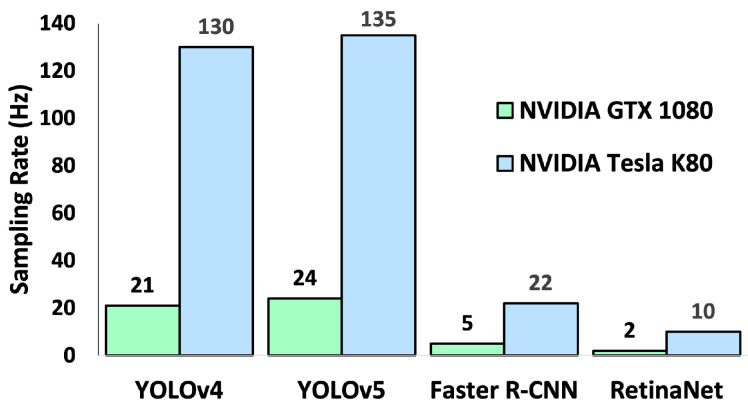
Sampling/inference rates in Hz are shown for two YOLO architectures, Faster R-CNN, and RetinaNet object detection networks.

**Figure 9 sensors-22-03240-f009:**
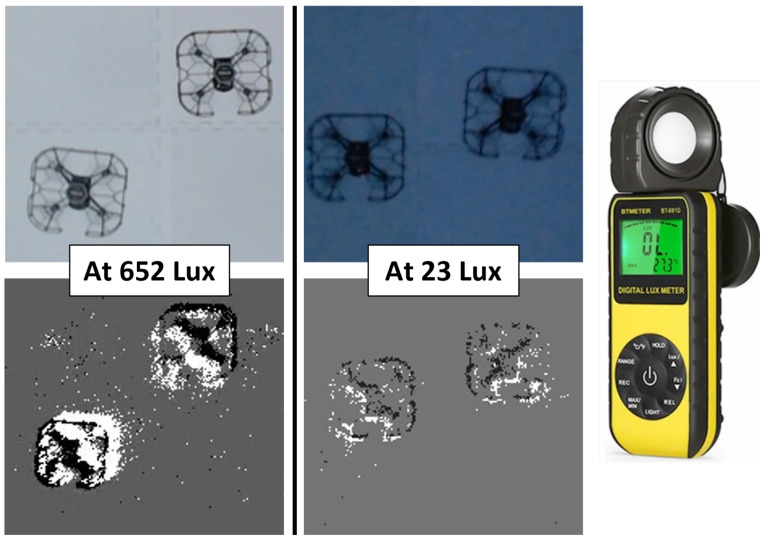
Detection performance was evaluated under various lighting conditions. LED light strips were used to modulate the brightness of the experiment environment. Also shown here is the Lux meter used for ambient light measurements.

**Figure 10 sensors-22-03240-f010:**
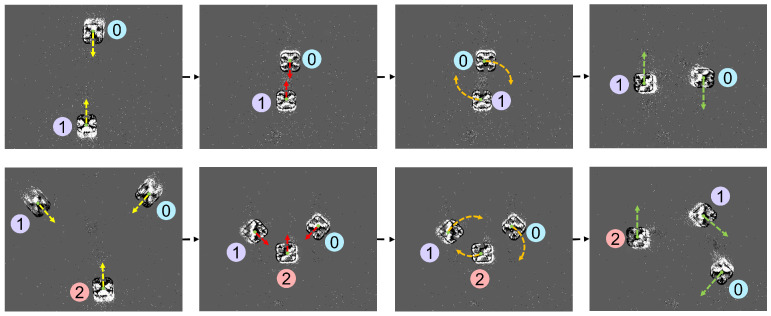
The motion capture system was used to test collision avoidance using a decentralized optimization-based motion planner. In each flight test, the quadrotors were able to successfully avoid $d_{safe}$ violations. Flight tests involved 2 quadrotors (**top** panel of images) and 3 quadrotors (**bottom** panel of images) flying toward each other. All quadrotors were operating between the speeds of 0.2 m/s and 1 m/s as determined by their respective motion planners.

**Figure 11 sensors-22-03240-f011:**
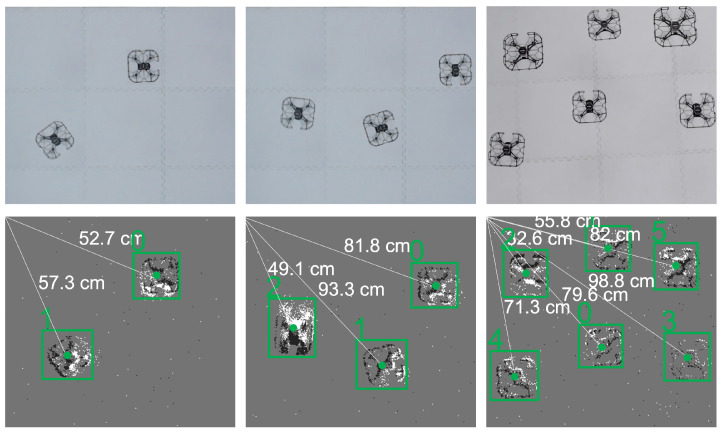
Indoor flight tests involving up to six quadrotors in the arena are depicted in RGB (**top**) and event (**bottom**) formats.

**Figure 12 sensors-22-03240-f012:**
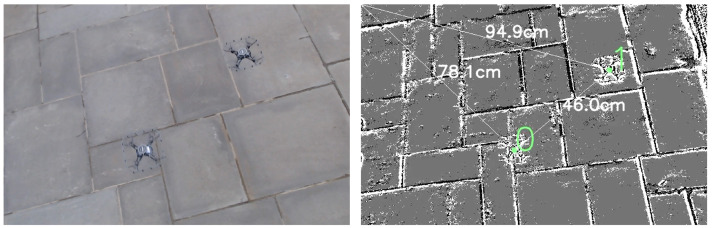
Experiments involving 2 quadrotors carried out in outdoor environments are depicted in RGB (**left**) and event (**right**) format. For the outdoor experiments, the RGB and event cameras were mounted at different locations on the imaging quadrotor. Due to the different camera perspectives, the quadrotors appear to be at slightly different positions in the RGB and event images shown here.

**Table 1 sensors-22-03240-t001:** Summary of the evaluations and relevant metrics performed in this work.

Evaluation Dimension	Evaluation Method	Evaluation Metric
NN Training Results	(Offline) Validation dataset	P, R, F
Performance comparison of NNs	(Offline) Testing Set	P, R, Sampling Rate
Ambient lighting conditions	Varied Lux levels	P, R
Unstable camera conditions (indoors)	Shaking fixed camera	P, R
Unstable camera conditions (outdoors)	Mounting camera facing downwards on a quadrotor	P, R
Effect of $n_{quad}$ on $T_{mission}$	Optimization-based motion planning	$T_{mission}$
Effect of $d_{safe}$ on $T_{mission}$	Optimization-based motion planning	$T_{mission}$
Effect of quadrotor paths on O	Optimization-based motion planning	O

**Table 2 sensors-22-03240-t002:** Average $T_{mission}$ (and rounded to zero decimals) in seconds for varying $d_{safe}$ and $n_{quad}$ for indoor experiments.

$d_{\mathrm{safe}}$ (cm)	$n_{\mathrm{quad}}$ = 2	$n_{\mathrm{quad}}$ = 3	$n_{\mathrm{quad}}$ = 6
$5\leq d_{safe}\leq 20$	12	14	N/A
$20\leq d_{safe}\leq 55$	11	12	19

**Table 3 sensors-22-03240-t003:** Boundary violation rate O results for each flight test path shape.

$n_{\mathrm{quad}}$	Square	Circle	Lawnmower	Spline
1—Indoor	0.04	0.08	0.08	0.07
2—Indoor	0.07	0.04	0.04	0.09
2—Outdoor	0.11	N/A	N/A	N/A

## Data Availability

Not applicable.

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
