# Peer review of "Event-Based Motion Capture System for Online Multi-Quadrotor Localization and Tracking"

_sensors, 2022, doi:10.3390/s22093240_

Round 1
Reviewer 1 Report
The paper deals with the positioning system for drones moving in the 3D space. Although I am not an expert in image processing and neural networks I think that the main contribution of the paper is focused on the design of a new low-cost localization and navigation system. However, I cannot see any new theoretical results. The authors takes advantage of well-known tools which are properly combined in order to obtain an original laboratory-scaled positioning system.
I believe that presented results are convincing for researchers interesting in various drones applications. In spite of that I think that the measurement quality of the proposed system should be compared with respect to other well recognized capture tracking applications such as OptiTrack. Currently it is hard to evaluate its properties only on the presented experimental scenarios.
I also think that the title of the paper is misleading as in my opinion the motion planning is not a subject of this work. I think that the manuscript should be focused on the one specific topic. Here, too many subjects, namely localization and navigation issues are taken into account.
Reviewer 2 Report
The authors of this paper provided a method to detect quadrotors via a camera and also provided a way to track them as well as plan their path. First, the work generated a first-of-its-kind data set for training the neural network, which is one of the main contributions of the manuscript. Following that the authors successfully trained and apply the YOLO neural network model to detect quad-rotors, plan their motion and track them. This paper contributes significantly to this area. The paper is well written and presented, except for some minor issues.
- The symbols, as well as the notations used in the paper, are redundant and hard to track. For instance t_a, e^t, t_a^n etc. It would be nice to use distinct notations. Futhermore, the notations should use different font styles for readability.
- The Experiment section is too long and needs to be streamlined.
- There is a recent performance study on the use of machine learning for quad-rotors motion planning. The authors are advised to look into it and include it in the citation
Jembre, Y.Z.; Nugroho, Y.W.; Khan, M.T.R.; Attique, M.; Paul, R.; Shah, S.H.A.; Kim, B. Evaluation of Reinforcement and Deep Learning Algorithms in Controlling Unmanned Aerial Vehicles. Appl. Sci. 2021, 11, 7240. https://doi.org/10.3390/app11167240
Reviewer 3 Report
This study introduces a relatively low-cost real-time motion-capture system for multi-quadrotor motion planning, in which event camera and well-performed deep learning network are utilized to implement. A decentralized motion planning algorithm is subsequently performed and successfully verified the effectiveness of this system. The structure of this article is clear. The topic in this paper is interesting and fits the scope of the journal. However, this article still needs some revisions before its acceptance and the corresponding comments are as given below:
- In Figure 3, why t=0.1 seconds shows the best results? The detailed discussion need to be added. What is the principle of parameter accumulation time?
- Can different environments with strong/weak intensity lighting represent the actual activities?
- The results showed that precision and recall values of YOLOv4 and YOLOv5 are similar enough. Since the YOLOv5 is faster, should it be regarded the first choice?
- As shown in figure 9, do all quadrotors fly at the same speed?
- How many motion planners are needed in the host machine?
- As shown in Figure 11, quadrotors that depicted in RGB and event format are not in the same position.
- As open-source models, the YOLOv5 and YOLOv4 are widely applied and discussed by other papers. Besides, the mentioned dataset has been presented in [17]. In this context, they cannot be regarded as the innovation or strengths of this paper.
Round 2
Reviewer 3 Report
Good work.
Author Response
Thank you for your encouraging feedback.